# Effect of Lauric vs. Oleic Acid-Enriched Diets on Leptin Autoparacrine Signalling in Male Mice

**DOI:** 10.3390/biomedicines10081864

**Published:** 2022-08-02

**Authors:** Jesús Fernández-Felipe, Adrián Plaza, Gema Domínguez, Javier Pérez-Castells, Victoria Cano, Francesco Cioni, Nuria Del Olmo, Mariano Ruiz-Gayo, Beatriz Merino

**Affiliations:** 1Department of Health and Pharmaceutical Sciences, Facultad de Farmacia, Universidad San Pablo-CEU, CEU Universities, 28660 Madrid, Spain; jesus.fernandezfelipe@ceu.es (J.F.-F.); adrian.plaza@imdea.org (A.P.); victoria.cano@ceu.es (V.C.); cioni.1694862@studenti.uniroma1.it (F.C.); 2Laboratory of Bioactive Products and Metabolic Syndrome (BIOPROMET), IMDEA Food Institute, 28049 Madrid, Spain; 3Department of Chemistry and Biochemistry, Facultad de Farmacia, Universidad CEU-San Pablo, CEU Universities, 28660 Madrid, Spain; gdommar@ceu.es (G.D.); jpercas@ceu.es (J.P.-C.); 4Departament of Psychobiology, Facultad de Psicología, Universidad Nacional de Educación a Distancia, 28040 Madrid, Spain; nuriadelolmo@psi.uned.es

**Keywords:** adipose tissue hypertrophy, leptin resistance, monounsaturated fatty acids, perigonadal adipose tissue, saturated fatty acids, subcutaneous adipose tissue, visceral adipose tissue

## Abstract

High-fat diets enriched with lauric acid (SOLF) do not enhance leptin production despite expanding white adipose tissue (WAT). Our study aimed at identifying the influence of SOLF vs. oleic acid-enriched diets (UOLF) on the autoparacrine effect of leptin and was carried out on eight-week-old mice consuming control chow, UOLF or SOLF. Phosphorylation of kinases integral to leptin receptor (LepR) signalling pathways (^705^Tyr-STAT3, ^473^Ser-Akt, ^172^Thr-AMPK), adipocyte-size distribution, fatty acid content, and gene expression were analyzed in WAT. SOLF enhanced basal levels of phosphorylated proteins but reduced the ability of leptin to enhance kinase phosphorylation. In contrast, UOLF failed to increase basal levels of phosphorylated proteins and did not modify the effect of leptin. Both SOLF and UOLF similarly affected adipocyte-size distribution, and the expression of genes related with adipogenesis and inflammation. WAT composition was different between groups, with SOLF samples mostly containing palmitic, myristic and lauric acids (>48% *w*/*w*) and UOLF WAT containing more than 80% (*w*/*w*) of oleic acid. In conclusion, SOLF appears to be more detrimental than UOLF to the autoparacrine leptin actions, which may have an impact on WAT inflammation. The effect of SOLF and UOLF on WAT composition may affect WAT biophysical properties, which are able to condition LepR signaling.

## 1. Introduction

White adipose tissue (WAT) plays a crucial role in body energy homeostasis due to its metabolic plasticity and endocrine function. Negative energy balance promotes lipolytic mechanisms leading to fatty acid (FA) and glycerol release, necessary to fulfil energy needs in other tissues; in contrast, during periods of overfeeding, WAT stores energy in the form of triglycerides (TG), useful for periods of food shortage. This mechanism contributes to limit ectopic accumulation of lipids in non-adipose tissues and reduces the risk of lipotoxicity [1,2]. Metabolic activity of WAT is regulated both by neuronal and endocrine signals, including adipokines [3,4,5], which are pivotal inorchestrating energy balance regulation by modulating food intake, WAT plasticity, and energy metabolism.

Long periods of elevated intake of simple carbohydrates and/or fat trigger adaptive remodelling of WAT, characterized by the increase in adipocyte size (hypertrophy) and proliferation (hyperplasia), affecting both subcutaneous (Sc-WAT) and visceral WAT (Vis-WAT). These changes are accompanied by a concomitant increase in leptin production [6,7]. In addition to its endocrine role, pivotal in regulating food intake and fat catabolism, physiological concentrations of leptin are endowed with an autocrine-paracrine function that promotes adipogenesis by a mechanism involving PPARγ [8]. In contrast, hyperleptinemia associated with diet-induced obesity (DIO) has been shown to inhibit adipocyte differentiation and to promote WAT dysfunction [3,9].

The detrimental effect of hyperleptinemia is associated with the development of leptin resistance in many tissues/organs [10,11,12,13,14,15] able to trigger WAT hypertrophy [16] and ectopic accumulation of lipids [17,18,19]. In fact, fat storage in adipocytes has been shown to require the inactivation of leptin paracrine activity [16]. Despite all this research demonstrating the consequences of hyperleptinemia and leptin resistance for WAT function [16,19,20,21], an adequate characterization of the WAT leptin receptor (LepR) functionality under conditions of DIO is lacking. Particularly, the specific influence that saturated vs. monounsaturated fat may have on WAT LepR responsiveness has not been sufficiently investigated. In the same vein, the effect of these types of fat on WAT morphology remains poorly characterized. It should be highlighted that diets enriched with saturated fat (SOLF), and particularly with lauric acid, do not enhance leptin production despite expanding WAT.

Our hypothesis is that saturated and monounsaturated fat differently affect the autoparacrine function of leptin. The aim of the current study is to investigate the influence of saturated and monounsaturated fat on LepR functionality in both Sc and Vis-WAT. For this purpose, we analysed the responsiveness of adipose LepR to exogenous leptin in mice that consumed (eight weeks) sugar-free diets enriched either in monounsaturated (high-oleic sunflower oil, UOLF) or saturated fat (palm kernel oil, SOLF). Moreover, we characterized the effect of these diets on WAT remodelling, which is known to be a leptin-sensitive process.

## 2. Material and Methods

### 2.1. Animals and Diets

Assays were carried out in C57BL male mice (Charles River, Écully, France) that started to consume the experimental diets on postnatal day 35 (p35). Experimental procedures were carried out in accordance with the European Union Laboratory Animal Care Rules (86/609/EEC) and were approved by the Animal Research Committee of San Pablo CEU University (PCD Ref. PROEX-017/18. Approval date, 23 March 2017). Four-week-old animals were single housed under a 12-h light/12-h dark cycle, in a temperature-controlled room (22 °C) with water and rodent chow available *ad libitum*. After one week, animals with similar average body weight (BW) were randomly assigned to one of three experimental groups that were fed (8 weeks) either standard chow (SD, Teklad global 2018, Harlan Laboratories, Indianapolis, IN, USA), SD enriched with 40% high-oleic sunflower oil (Unsaturated OiL-enriched Food, UOLF), or SD enriched with 40% palm kernel oil (Saturated OiL-enriched Food, SOLF) (see diets’ composition in Appendix A). UOLF and SOLF provided about 70% energy from fat (vs. 18% in chow), with oleic acid (18.1 cis 9; 31%) and lauric acid (12:0)/palmitic acid (16:0) (20%/6%) being the most abundant FA in UOLF and SOLF, respectively. BW was monitored once a week. WAT morphology and composition, and gene expression were analysed in samples obtained from animals used in a previous investigation [12]. A second group of mice was employed to assess leptin resistance in WAT. Group sizes (6–7 animals per group) were calculated based on previous studies by our group and an expected 10% increase of BW after 8 weeks of dietary treatment, a statistical power >85% and *p* < 0.05.

### 2.2. Determination of Subcutaneous and Visceral White Adipose Tissue Composition

Lipid composition was determined by means of ^13^C nuclear magnetic resonance (^13^C-NMR) of an FA pool obtained by saponification of total WAT lipids. For saponification, total Sc or Vis-WAT lipids, extracted by using the Folch method modified by Herrera and Ayanz [22], 1972 (100–130 mg), were dissolved in 0.7 mL of tetrahydrofuran and mixed with 0.7 mL of 1 M NaOH. The resulting mixture was stirred at 50 °C for 48 h, then diluted with 2 mL of MQ water and acidified with 1 M HCl until pH = 1. The mixture was extracted twice with 4 mL of CH_2_Cl_2_, washed with 2 mL of brine, dried over MgSO_4_ and filtered. Upon elimination of the organic solvent, colourless oily mixtures of FAs were isolated.

For ^13^C-NMR spectra recording, 100 mg of each FA mixture sample were dissolved in 1.4 mL of deuterated chloroform (CDCl_3_) and 0.5 mL of this solution was placed in NMR tubes. Spectra were recorded on a Bruker AM-400 MHz spectrometer, operating at 100.6 MHz, at 305 K. Chemical shifts (δ) are expressed in ppm from the central signal of CDCl_3_ (77.0 ppm). Before Fourier transformation, exponential multiplication with a LB = −0.7 was applied. Spectra were acquired using an inverse gated decoupling sequence so that fully decoupled spectra with no NOE were recorded. A total of 128 scans with a delay between scans of 60 s (>8 times T1) was used. Our analysis was centred on the methylene carbon resonances around 20–35 ppm [23].

For the integration, well-resolved signals of each acid were used. This gave a molar% average which was transformed into mass% average. However, the presence of palmitic and myristic acids in SOLF samples was observed by the splitting of the signals at 29.355 and 31.927 ppm. Unfortunately, this splitting does not give completely resolved signals. Therefore, the weight of these two acids was assigned higher error, although their combined contribution has the same general error of 5%.

### 2.3. Hematoxylin/Eosin Staining and Quantification of Adipocyte Size

WAT samples were fixed in 4% formaldehyde for 10 days, then washed with water (2 h), dehydrated with ethanol and subsequently embedded in paraffin. Thin serial sections (5 μm) were obtained with a vertical rotary microtome (Leica RM 2125RT) and mounted in glass-slides. Slices were stained with hematoxylin/eosin (H/E; Thermo Scientific, Madrid, Spain) to assess cellular morphology and diameter quantification. Two vision fields per slice were randomly selected and quantified. After staining, sections were dehydrated in ethanol and xylene and mounted with DPX. Samples were directly observed (20×) by using an Eclipse 50i-Nikon microscope equipped with a camera (DS-5M) and NIS-Elements software. Adipocyte-size classification was made according to the criteria provided by Verboven et al., 2018 [24].

### 2.4. Leptin Resistance Assessment in White Adipose Tissue

To evaluate whether leptin signalling in WAT was modified by the dietary treatment, an experiment aimed at evaluating the effect of exogenous leptin on LepR responsiveness was carried out after 8 weeks of dietary treatment. Mice received (i.p.) either saline or 1 mg/kg mouse recombinant leptin (Sigma, St. Louis, MO, USA) at 0900 h. After 60 min, animals were decapitated under isoflurane anaesthesia, blood was collected in chilled EDTA-coated polypropylene tubes, and Sc and Vis-WAT were dissected and stored at −80 °C until assay. Both tissues were prepared for Western blot, as described below. The dose of leptin was chosen based on previous studies of our group [11]. This dose has been shown to provide plasma leptin levels of approx. 80 ng/mL [11], which fits into the range of severe hyperleptinemia.

### 2.5. Western Blotting

For Western blotting, tissues were homogenized in ice-cold buffer containing 30 mM HEPES (pH = 7.9), 0.1 mM Na_4_P_2_O_7_, 5 mM EDTA, 1% triton X-100, 0.5% glycerol, 1 μg/mL aprotinin, 1 μg/mL leupeptin, 1 mM sodium fluoride, 7.5 mM trisodium orthovanadate. Tubes containing homogenates were frozen at −80 °C and thawed at 37 C three consecutive times, then centrifuged for 10 min at 4 °C. Equivalent amounts of proteins (50 μg) present in the supernatant were loaded in Laemli buffer (50 mM Tris, pH = 6.8, 10% sodium dodecyl sulphate, 10% glycerol, 5% mercaptoethanol, and 2 mg/mL blue bromophenol) and size-separated in 7% SDS-PAGE. Proteins were transferred to nitrocellulose membranes (GE Healthcare, Little Chalfont, Buckinghamshire, UK) using a transblot apparatus (Bio-Rad, Hercules, CA, USA). For immunoblotting, membranes were blocked with 5% non-fat dried milk in Tween-PBS for 1 h. Primary antibodies (Table 1) were applied at the convenient dilution, overnight at 4 °C. After washing, appropriate secondary antibodies (anti-rabbit, IgG-peroxidase conjugated) were applied for 1 h at a dilution of 1:2000.

Blots were washed, incubated in enhanced chemiluminescence reagent (ECL Prime; GE Healthcare, Chicago, IL, USA), and bands were detected using the ChemiDoc XRS+ Imaging System (BioRad, Hercules, CA, USA). To ensure equal loading of samples, blots were re-incubated with β-actin monoclonal antibody (Affinity Bioreagents, Ancaster, CO, USA). Blots were detected using the ChemiDoc XRS+ Imaging System (Bio-Rad, Hercules, CA, USA). Values for pSTAT3, pAkt and pAMPK were normalized with STAT3, Akt and AMPK, respectively.

### 2.6. RNA Preparation and Quantitative Real-Time PCR

Total RNA from Sc and Vis-WAT was extracted by using the Tri-Reagent protocol (Sigma, St. Louis, MO, USA). cDNA was then synthesized from 1 μg total mRNA by using a high-capacity cDNA RT kit (Bio-Rad, Hercules, CA, USA). Quantitative RT-PCR was performed by using designed primer pairs (Integrated DNA Technologies, Coralville, IA, USA. Table 2). SsoAdvanced Universal SYBR Green Supermix (Bio-Rad, Hercules, CA, USA) was used for amplification according to the manufacturer’s protocols, in CFX96 Real Time System (Bio-Rad, Hercules, CA, USA). Values were normalized to the housekeeping gene *Actb* and *18s*. The ∆∆C(T) method was used to determine relative expression levels. Statistics were performed using ∆∆C(T) values [25]. All samples were run in duplicate.

### 2.7. Statistics

Adipocyte size distribution and diet-effect over leptin signalling were analysed by two-way ANOVA with a post-hoc Bonferroni correction. Individual effects were analysed by one-way ANOVA followed by a Bonferroni post-hoc analysis. Data were expressed as mean ± S.E.M. and statistical significance was set at *p* < 0.05. Normal distribution and variance homogeneity were assessed by means of the Bartlett and Brown–Forsyte test. Outliers were identified using the ROUT method (Q = 1%). All statistics were performed using GraphPad Prism software (GraphPad Software Inc. San Diego, CA, USA; Version 7.0a).

## 3. Results

### 3.1. UOLF, but Not SOLF, Preserved the Effect of Exogenous Leptin on STAT3, Akt and AMPK Phosphorylation Both in Subcutaneous and Visceral WAT

The effect of dietary treatment and acute leptin administration (1 mg/kg) on ^705^Tyr-STAT3, ^473^Ser-Akt and ^172^Tyr-AMPK phosphorylation levels was evaluated both in Sc and Vis-WAT.

Subcutaneous WAT: As illustrated in Figure 1A–C, two-way ANOVA (see F values in Table 3) revealed an effect of leptin on relative pSTAT3 (*p* < 0.001), pAkt (*p* < 0.001), and pAMPK levels (*p* < 0.001), that was dependent on the type of diet consumed (*p* < 0.05, *p* < 0.05 and *p* < 0.01, respectively). Post-hoc analyses indicated that leptin increased pSTAT3, pAkt and pAMPK in control and UOLF mice but not in SOLF animals (see complete post-hoc analyses in Appendix A).

Visceral WAT: In this tissue, a significant effect of leptin on pSTAT3 (see F values in Table 4) (*p* < 0.001), pAkt (*p* < 0.001) and pAMPK relative levels (*p* < 0.001) was also detected. The effect of leptin depended on the type of diet consumed in the case of pSTAT3 (*p* < 0.05) and pAkt (*p* < 0.01) but not for pAMPK. Further post-hoc analysis revealed an effect of leptin on pSTAT3 and pAkt in SD and UOLF animals (see complete post-hoc analyses in Appendix A).

### 3.2. SOLF and UOLF Increased Adipocyte Size Both in Subcutaneous and Visceral WAT

To evaluate the influence of dietary treatment with UOLF and SOLF on adipocyte size and number, Sc and Vis-WAT sections were stained with H/E. As illustrated in Figure 2A,C, H/E staining revealed that the average size of Sc adipocytes was larger in UOLF samples than in their corresponding controls (one-way ANOVA; F_(2,19)_ = 4.02, *p* < 0.05), while no effect of SOLF was detected. In the case of Vis-WAT (Figure 2B,D), the adipocyte size appeared also to be significantly greater in UOLF-treated mice than in control and SOLF-treated mice (one-way ANOVA; F_(2,19)_ = 26.51, *p* < 0.001), although SOLF adipocytes were also hypertrophic (*p* < 0.05).

Adipocyte-size distribution curves (Figure 2E,F) were also analysed. Two-way ANOVA revealed a significant influence of dietary treatment on adipocyte size distribution both in Sc-WAT (Figure 2E; F_(10,246)_ = 3.243, *p* < 0.001) and Vis-WAT (Figure 2F; F_(10,246)_ = 19.64, *p* < 0.001). Further one-way ANOVA indicated that the dietary treatment reduced the proportion of small adipocytes (0–1000 µm^2^; Figure 1G) in Sc-WAT (F_(2,19)_ = 3.497, *p* < 0.05) and enhanced that of medium (1000–4000 µm^2^; F_(2,19)_ = 2.799; *p* = 0.07) and large adipocytes (>4000 µm^2^; F_(2,19)_ = 2.690; *p* = 0.09). In the case of Vis-WAT, small adipocytes (Figure 2H) were also reduced by the diets (F_(2,19)_ = 34.18, *p* < 0.001), while medium (F_(2,19)_ = 15.33, *p* < 0.001) and large adipocytes (F_(2,19)_ = 19.98, *p* < 0.001) were more abundant in these groups. It must be highlighted that in Vis-WAT, adipocyte-size distribution was different between SOLF and UOLF with small adipocytes being more abundant in SOLF samples (*p* < 0.05), and large adipocytes being more abundant in UOLF (*p* < 0.01).

### 3.3. UOLF and SOLF Inhibited the Expression of Genes Involved in White Adipocyte Differentiation but Neither the Expression of Pro-Inflammatory and Pro-Fibrotic Factors nor Leptin Receptor Genes Were Affected

The expression of peroxisome proliferator-activated receptor gamma (*Pparg*) and CCAAT enhancer binding protein α (*Cebpa*) genes was analysed both in Sc and Vis-WAT. In the Sc-WAT, UOLF and SOLF treatment failed to modulate *Cebpa* and *Pparg* gene expression (Figure 3A). In contrast, in the Vis-WAT (Figure 3B) both *Cebpa* (F_(2,12)_ = 52.06, *p* < 0.001) and *Pparg* (F_(2,19)_ = 12.96, *p* < 0.001) expression was repressed by SOLF and UOLF.

Both diets were devoid of effect on the expression of the inflammation-related genes *Il1*, *Il6* and *Tnfa* both in Sc (Figure 3C) and Vis-WAT (Figure 3D), although the expression levels of *Tnfa* were at the limit of statistical significance. Similarly, SOLF and UOLF were without effect on gene expression of the pro-fibrotic gene *Col1a1*. An effect of SOLF was identified in the case of *Col3a1* in the Sc-WAT (F_(2,16)_ = 6.483, *p* < 0.01) (Figure 3 E,G).

The expression of the leptin receptor (*Lepr*) remained unchanged both in Sc and Vis-WAT (Figure 3F,H).

### 3.4. SOLF and UOLF Differently Affect the Composition of Subcutaneous and Visceral WAT

As appears summarized in Table 5, SOLF and UOLF treatment differently affected the composition of both Vis and Sc-WAT. In the case of Vis-WAT, UOLF treatment triggered a 40% increase of oleic acid content to the detriment of palmitic acid. In contrast, SOLF treatment barely modified the proportion of oleic acid but promoted the accumulation of both myristic (25%) and lauric (22%) acids, which were absent in control and UOLF Vis-WAT.

With regard to Sc-WAT composition, UOLF treatment also triggered a dramatic increase in oleic acid content (46% increment) to the detriment of palmitic acid (23.5% decrease) and linoleic acid (22% decrease). Treatment with SOLF led to a small decrease in both palmitic (10%) and linoleic acid content (17%), and promoted the storage of myristic and lauric acids, which were also absent in control and UOLF tissues.

## 4. Discussion

Leptin drives the crosstalk between WAT and organs/tissues regulating energy balance and is involved in the endocrine and autoparacrine control of energy metabolism [26,27,28]. In the current study, we have shown that long-term consumption of diets enriched in saturated FA (SOLF) affects the phosphorylation levels of protein kinases integral to LepR signalling pathways and LepR responsiveness to leptin in WAT. In contrast, unsaturated diets (UOLF) were devoid of these effects. Nevertheless, both SOLF and UOLF had a significant impact on WAT remodelling.

One of the most remarkable set of data in our study is the increase in basal pSTAT3, pAkt and pAMPK observed specifically in SOLF mice, particularly in the Sc-WAT. This is an interesting point since we previously reported that WAT leptin expressions as well as plasma leptin levels are higher in UOLF than in SOLF mice [12,29] and, therefore, one would expect the activation of these signalling kinases in UOLF rather than in SOLF animals. We speculate that this effect may be integral to an adaptive response aimed at increasing adipogenesis and/or lipogenesis in Sc-WAT, since STAT3, Akt and AMPK have been shown to regulate a substantial number of substrates involved in these processes [8,30,31]. Notably, STAT3 is abundantly expressed in pre-adipocytes [32] and pSTAT3 has been shown to contribute, at least in vitro, to adipocyte differentiation [33]. The role of Akt has been better established as it is involved both in the lipogenic and adipogenic effects of insulin [34]. With regard to AMPK, the increase in phosphorylated ^172^Tyr-AMPK elicited by SOLF is coherent with the inhibition of lipogenic pathways triggered by HFDs and also with the prominent role of AMPK on adipocyte proliferation [35,36]. This finding suggests that the endocrine environment generated by SOLF has consequences for signalling pathways relevant for Sc-WAT adaptation to the metabolic challenge evoked by an excess of saturated fat. In this context, the apparent lack of effect of leptin on STAT3/Akt/AMPK phosphorylation observed in SOLF mice is of difficult interpretation since leptin effects could be masked by the above-mentioned diet effect. In contrast, in the Vis-WAT, SOLF only increased basal phosphorylation levels of Akt, although a desensitization of LepR to leptin effects was globally observed in this tissue. With respect to UOLF effects, it is noteworthy that the phosphorylation levels of STAT3, Akt and AMPK were not affected by the diet itself although they were enhanced by leptin in a similar way that leptin did in animals consuming regular chow. This circumstance was observed both in Sc and Vis-WAT, demonstrating that unsaturated diets fully preserve LepR responsiveness in WAT and indicating that the autoparacrine effect of leptin is not affected by these diets. In any case, differences in LepR sensitivity between dietary treatments are not linked to different levels of LepR density since no change was detected in the expression of the *Lepr*.

Altogether, the results regarding kinase phosphorylation show that SOLF has a greater impact than UOLF on the autoparacrine effect of leptin. On the basis of previous studies, showing that the rise of leptin levels is more prominent in UOLF than in SOLF-treated mice [12,29], our current findings suggest the involvement of other factors than leptin on STAT3/Akt/AMPK signalling dysregulation. In this line, inflammation evoked by saturated FA could contribute to SOLF effects [37]. Nevertheless, although our data concerning the impact of SOLF/UOLF on WAT inflammation are restricted to the quantification of *Il6*, *Il1* and *Tnfa* mRNAs, they suggest that local inflammatory mediators do not play a major role in SOLF effects. In the same way, fibrosis does not appear to be a prominent hallmark of SOLF and UOLF WAT since the expression of the fibrosis markers *Col1a1* and *Col3a1* remained unchanged in Vis-WAT and only an increase in *Col3a1* was detected in the Sc-WAT of SOLF mice. Such an increase may be eventually related to actin re-organization required for adipogenesis [38]. Finally, some studies point to a negative effect of excessive adipocyte FA uptake on LepR functionality [39]; related to that, palmitic acid has been shown to impair LepR signalling within the hypothalamus [40], a finding that would point to a particular effect of saturated FA on LepR signalling. It is noteworthy that both SOLF and UOLF contain palmitic acid, while lauric acid is only present in SOLF. Although lauric acid is currently considered as a FA with less impact than palmitic acid on WAT inflammation [41], our results indicate that lauric acid does not prevent the harmful consequences of saturated fat intake. Related to that, our findings suggest that human dietary habits including foods enriched in lauric acid (i.e., coconut oil) should be analysed with a view to properly evaluating their eventual health risks.

Taken together with the morphological data, the effect of SOLF/UOLF on LepR signalling points to an adaptive remodelling in Sc-WAT. In fact, although adipocyte size distribution was similar in SOLF, UOLF and control Sc-WAT, both SOLF and UOLF mice displayed larger fat depots than control animals [12], allowing us to conclude that the adipocyte number in Sc-WAT is larger in SOLF/UOLF than in control mice. This supports the existence of active adipogenesis, a circumstance that is coherent with the basal increase of phosphorylated kinases identified in SOLF mice as well as in UOLF-treated mice receiving an acute leptin challenge.

In contrast, both diets decreased the proportion of small adipocytes and enhanced that of large adipocytes in Vis-WAT, leading us to hypothesize that SOLF/UOLF similarly limit adipogenesis in this tissue. This hypothesis is corroborated by the down-regulation in the expression of the adipogenic genes, *Cebpa* and *Pparg* [42], which remained unmodified in Sc-WAT. Collectively, our data indicate that both SOLF and UOLF promote adipocyte hypertrophy but inhibit adipogenesis in Vis-WAT. Despite the lack of sufficient data regarding the presence of inflammation and/or fibrosis, we speculate that 8-week SOLF/UOLF intake leads to an incipient pathologic remodelling of Vis-WAT [43]. In any case, further studies aimed at characterizing the expression of inflammatory markers in the isolated stromal fraction of WAT would be necessary for a proper characterization of this issue. Similar considerations can be made regarding fibrosis [39].

Otherwise, we have identified significant differences between SOLF, UOLF and control mice regarding WAT composition. We want to highlight that WAT composition was analysed by means of ^13^C-NMR, which is less performant than gas chromatography for this kind of study. In any case, ^13^C-NMR is also considered a reliable method to analyse lipids in complex mixtures [44]. Thus, compared to control Vis-WAT [unsaturated FA]/[saturated FA], ratios were moderately decreased in SOLF (0.6 vs. 0.8) and enhanced in UOLF animals (7.2 vs. 0.8). This pattern, which was also observed in the Sc-WAT (1.1 vs. 2.0, SOLF vs. controls; 11.4 vs. 2.0, UOLF vs. controls), is coherent with diet compositions and introduces an interesting point of discussion, related to the influence that the mechanical properties of WAT may have on WAT functionality [45]. Thus, despite the lack of proper biophysical data, one could hypothesize that UOLF WAT would be more fluid than SOLF and control WATs. In fact, melting points of TGs containing oleyl and linoleyl moieties (which presumably are major components of UOLF Vis and Sc-WAT), would be around 5 °C, and therefore liquid at mouse body temperature, while TGs contained in control and SOLF WAT should be solid at this temperature. Reasoning in this way, SOLF WAT TGs should present higher melting points than control WAT TGs. With all that in mind, variations in WAT rigidity may account for some of the differences detected between SOLF and UOLF, regarding size adipocyte distribution and LepR signalling. In fact, adipogenesis is a mechanosensitive process, and increasing evidence links lipid production in adipocytes to a particular mechanical environment [46]. This circumstance, together with the differential effect of saturated and unsaturated FA on adipocyte maturation [47], allows us to speculate that differences between SOLF and UOLF mice regarding *Lep* gene expression previously reported by us [12,29] may be linked to the biophysical properties of the tissue [45,48,49].

Similar considerations could be made to interpret differences in LepR signalling. Interestingly, membrane protein conformation and activity have been shown to depend on the membrane lipid environment [50], which may be theoretically modified by SOLF/UOLF [51]. Specifically, membrane cell composition and fluidity have been shown to have a direct impact on membrane receptor functionality [52,53].

Another interesting point of our study deals with the reduction in linoleic acid content evoked both by SOLF and UOLF in the Sc-WAT. Since Ω-6 PUFA have been shown to promote inflammation [54], reduction in linoleic acid content by SOLF/UOLF may balance other detrimental effects elicited by these diets, and explain why Sc-WAT is less sensitive to SOLF/UOLF effects than Vis-WAT. In any case, further experiments need to be carried out to confirm the differential effects of saturated vs. unsaturated fats on mechanical properties/stiffness and fluidity of white adipocyte membranes.

It has to be noted that our study was carried out only on male mice and, therefore, complementary studies carried out on females are necessary to further characterize sex-dependent differences. With regard to the eventual translation of our results to humans, the finding that SOLF triggers a similar increase in adiposity to UOLF but causes a deficit in leptin production and impairs LepR signalling supports the concept that an excess of lauric acid (i.e., coconut oil) may be detrimental for leptin effects and therefore for WAT function.

In summary, despite both SOLF and UOLF having a similar effect on WAT expansion, these diets have a different impact on LepR signalling within Sc and Vis-WAT, with SOLF-treated mice displaying apparent leptin resistance, particularly in the Vis-WAT. Our findings show that the intake of food enriched with either oleic (UOLF) or lauric/palmitic acids (SOLF) may differently affect the autoparacrine role of leptin. Overall, our findings suggest that SOLF desensitize signalling pathways downstream of LepR in adipocytes, which could limit WAT adaptive mechanisms. Nevertheless, and despite the fact that UOLF does not alter LepR signalling, this diet triggers adipocyte hypertrophy to a similar extent to SOLF, indicating that, regardless of diet composition, an excessive intake of fat is harmful to WAT function. Further research aimed at correlating LepR responsiveness with adipocyte function in SOLF- and UOLF-fed mice appears necessary.

## Figures and Tables

**Figure 1 biomedicines-10-01864-f001:**
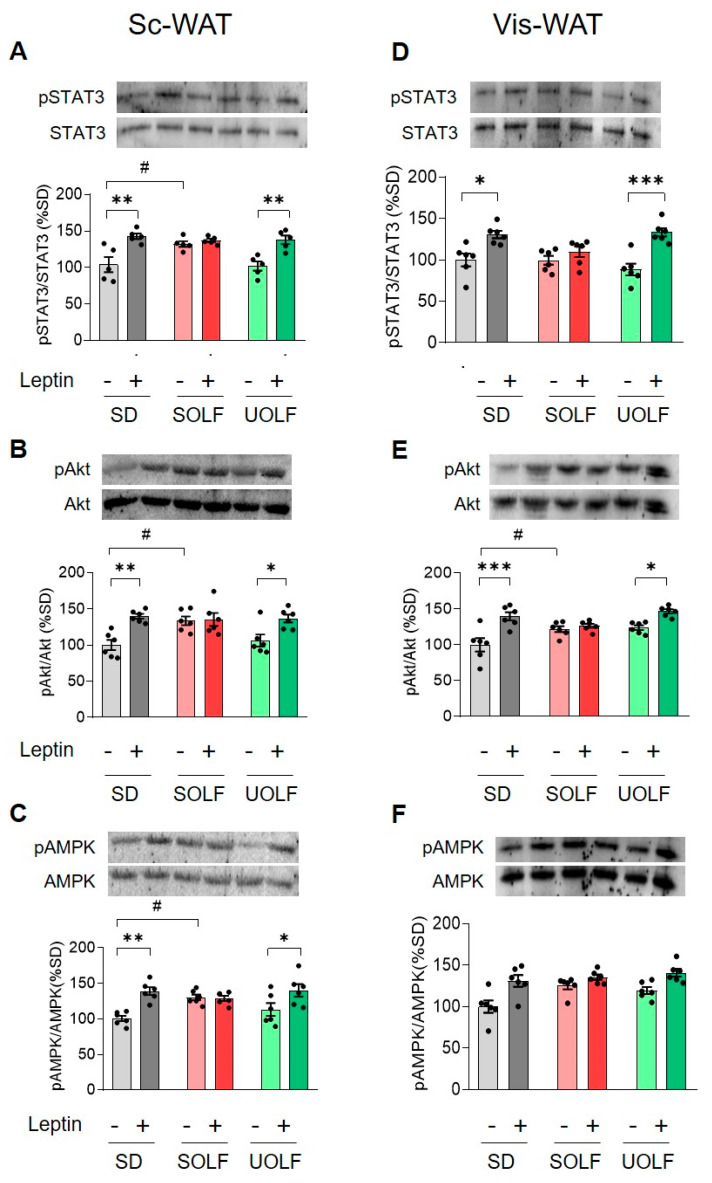
Influence of dietary treatment and acute leptin administration on phosphorylation levels of Tyr^705^-STAT3, Ser^473^-Akt and Tyr^172^-AMPK. Graphs show representative blots and relative levels of phosphorylated STAT3, Akt and AMPK in Sc-WAT (**A**–**C**) and Vis-WAT (**D**–**F**). Values are means ± S.E.M. (*n* = 6 per group). * *p* < 0.05, ** *p* < 0.01 and *** *p* < 0.01, compared to the corresponding saline group; ^#^ *p* < 0.05 compared to the saline-SD group (Bonferroni test).

**Figure 2 biomedicines-10-01864-f002:**
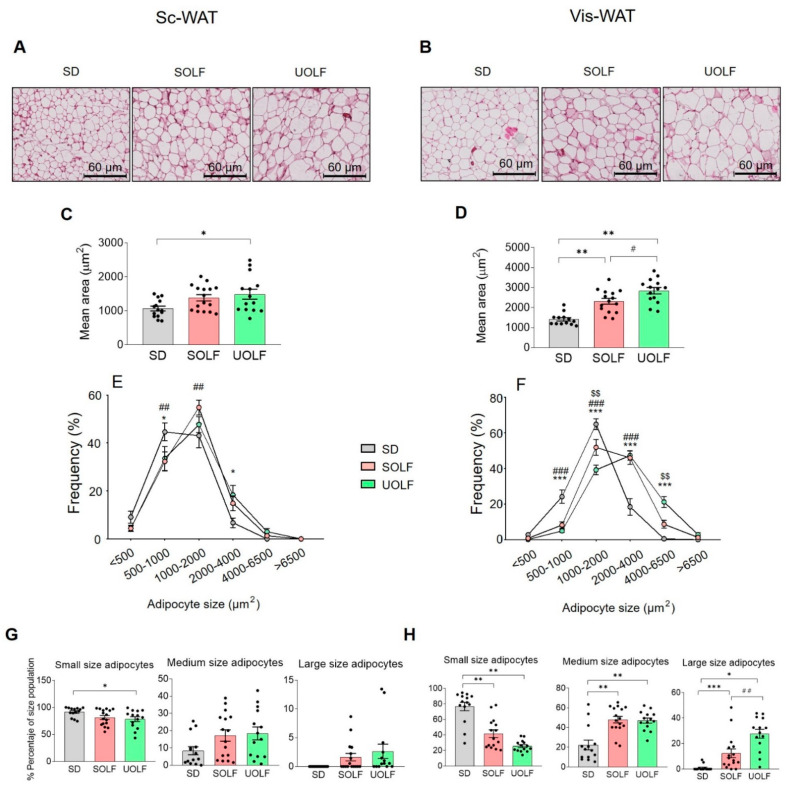
Effect of UOLF and SOLF diet on adipocyte morphology. Representative photomicrographs (×20) of H/E stained Sc (**A**) and Vis-WAT (**B**) from mice that consumed either SD, UOLF or SOLF. (**C**,**D**) graph bars show adipocyte mean areas in Sc and Vis-WAT, respectively. Graphs (**E**,**F**) illustrate adipocyte size distribution of Sc and Vis-WAT, respectively. Graphs (**G**,**H**) show the percentage of small, medium, and large adipocytes in Sc and Vis-WAT, respectively. Values are means ± S.E.M. (*n* = 6–7 per group). * *p* < 0.05, ** *p* < 0.01, *** *p* < 0.001 compared to SD. ^#^ *p* < 0.05 and ^##^ *p* < 0.01 compared to SOLF (Bonferroni test).

**Figure 3 biomedicines-10-01864-f003:**
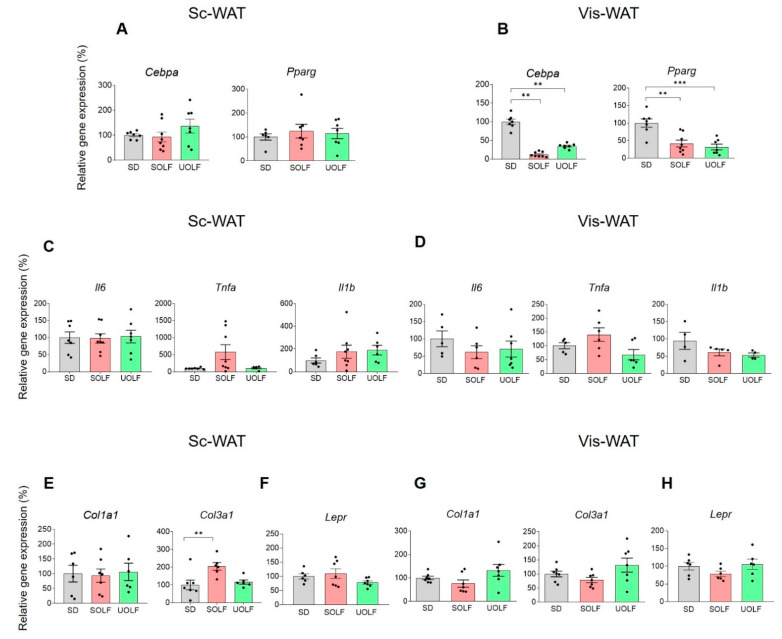
Effect of UOLF and SOLF on expression levels of the adipogenic genes *Pparg* and *Cebpa* in Sc and Vis-WAT. Figure (**A**) shows the lack of effect of SOLF and UOLF on mRNA levels of *Cebpa* and *Pparg* in Sc-WAT. In contrast, in Vis-WAT, both SOLF and UOLF down-regulated the expression of these genes (**B**). Figures (**C**,**D**) show mRNA levels of *Il1*, *Il6* and *Tnfa* in Sc and Vis-WAT, respectively. Figures (**E**,**G**) show the expression levels of *Col1a1* and *Col3a1* in Sc and Vis-WAT, respectively. Figures (**F**,**H**) show the lack of effect of SOLF and UOLF on expression levels of *Lepr* in Sc and Vis-WAT, respectively. Values are means ± S.E.M. (*n* = 6–7 per group). ** *p* < 0.01, *** *p* < 0.001, compared to their respective controls (Bonferroni test).

**Table 1 biomedicines-10-01864-t001:** Primary antibodies used in Western blot.

Antigen	Manufacturer	Ref.	Host and Molecular Weight	Dilution
STAT3	Santa Cruz Biotechnology	Sc-483	Rabbit polyclonal, 86 kDa	1/250
pSTAT3 (Y705)	Cell Signalling	#9131	Rabbit polyclonal, 86 kDa	1/250
Akt	Cell Signalling	#9272	Rabbit polyclonal, 60 kDa	1/500
pAkt (S473)	Cell Signalling	#9271	Rabbit polyclonal, 60 kDa	1/500
AMPK	Cell Signalling	#2532	Rabbit polyclonal, 62 kDa	1/500
pAMPK (T172)	Cell Signalling	#2531	Rabbit polyclonal, 62 kDa	1/500
Anti-rabbit IgG	Santa Cruz Biotechnology	Sc-2357	Mouse monoclonal, HRP	1/2000

**Table 2 biomedicines-10-01864-t002:** Designed primer pairs used in this study.

mRNA	Forward (5′→3′)	Reverse (5′→3′)
*Cebpa*	CCGATGAGCAGTCACCTCC	AGGAACTCGTCGTTGAAGGC
*Col1a1*	ACTGCCCTCCTGACGCAT	AGAAAGCACAGCACTCGCC
*Col3a1*	CCCATGACTGTCCCACGTAAGCAC	TGGCCTGATCCATATAGGCAATACTG
*Il1b*	TGCCACCTTTTGACAGTGATG	TGATACTGCCTGCCTGAAGC
*Il6*	TACCACTTCACAAGTCGGAGGC	CTGCAAGTGCATCATCGTTGTTC
*Pparg*	CATGGTTGACACAGAGATGCCATTCTG	TTGATCGCACTTTGGTATTCTTGGAGC
*Tnfa*	GGTGCCTATGTCTCAGCCTC	GCTCCTCCACTTGGTGGTTT
*Lepr*	GCAGCAAAAGGAAGCATTGGA	GGTGAGGAGCAAGAGACTGG
*18s*	GGGAGCCTGAGAAACGGC	GGGTCGGGAGTGGGTAATTT
*Actb*	TGGTGGGAATGGGTCAGAAGGACTC	CATGGCTGGGGTGTTGAAGGTCTCA

**Table 3 biomedicines-10-01864-t003:** F values obtained from two-way ANOVA to determine the effect of leptin and dietary treatment on ^705^Tyr-STAT3, ^473^Ser-AkT and ^172^Tyr-AMPK phosphorylation in Sc-WAT.

	F Value		
Protein	Leptin Effect	Diet Effect	Interaction Leptin × Diet
pSTAT3/STAT3	F_(1,30)_ = 42.89; *p* < 0.001	F_(2,30)_ = 6.890; *p* < 0.01	F_(2,30)_ = 4.393; *p* < 0.05
pAkt/Akt	F_(1,30)_ = 18.92; *p* < 0.001	F_(2,30)_ = 2.785; *p* = 0.07	F_(2,30)_ = 4.214; *p* < 0.05
pAMPK/AMPK	F_(1,30)_ = 17.86; *p* < 0.001	F_(2,30)_ = 1.321; n.s.	F_(2,30)_ = 5.466; *p* < 0.01

**Table 4 biomedicines-10-01864-t004:** F values obtained from two-way ANOVA to determine the effect of leptin and dietary treatment on^705^Tyr-STAT3, ^473^Ser-pAkt and ^172^Tyr-AMPK phosphorylation in Vis-WAT.

	F Value		
Protein	Leptin Effect	Diet Effect	Interaction Leptin × Diet
pSTAT3/STAT3	F_(1,30)_ = 32.93; *p* < 0.001	F_(2,30)_ = 1.558; n.s.	F_(2,30)_ = 4.114; *p* < 0.05
pAkt/Akt	F_(1,30)_ = 27.75; *p* < 0.001	F_(2,30)_ = 4.705; *p* < 0.05	F_(2,30)_ = 5.727; *p* < 0.01
pAMPK/AMPK	F_(1,30)_ = 22.64; *p* < 0.001	F_(2,30)_ = 4.996; *p* < 0.05	F_(2,30)_ = 1.914; n.s.

**Table 5 biomedicines-10-01864-t005:** Fatty acid composition of Vis- and Sc-WAT.

	Vis-WAT					
	Standard Diet		SOLF		UOLF	
FA	Mol%	Mass%	Mol%	Mass%	Mol%	Mass%
Oleic	45.4 ± 2.3	47.9 ± 2.4	36.5 ± 1.8	42.3 ± 2.1	87.7 ± 4.4	89.0 ± 4.4
Palmitic	54.5 ± 2.7	52.1 ± 2.6	10.2 ± 0.5	10.8 ± 0.5	12.2 ± 0.6	11.0 ± 0.5
Myristic	ND	ND	26.6 ± 1.3	25.0 ± 1.2	ND	ND
Lauric	ND	ND	26.6 ± 1.3	21.9 ± 1.1	ND	ND
Linoleic	ND	ND	ND	ND	ND	ND
	**Sc-WAT**					
Oleic	35.6 ± 1.8	36.8 ± 1.8	37.7 ± 1.9	41.6 ± 2.1	81.9 ± 4.1	82.7 ± 4.1
Palmitic	33.1 ± 1.6	31.1 ± 1.5	20.0 ± 1.0	20.0 ± 1.0	8.1 ± 0.4	7.5 ± 0.4
Myristic	ND	ND	9.8 ± 0.5	8.7 ± 0.4	ND	ND
Lauric	ND	ND	18.5 ± 0.9	14.4 ± 0.7	ND	ND
Linoleic	31.3 ± 1.6	32.1 ± 1.6	13.9 ± 0.7	15.5 ± 0.8	9.8 ± 0.5	9.8 ± 0.5

Data are means ± S.E.M. of 3–4 samples/group. ND, not detected.

## Data Availability

Data is contained within the article.

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
