# Peer review of "Effect of Lauric vs. Oleic Acid-Enriched Diets on Leptin Autoparacrine Signalling in Male Mice"

_biomedicines, 2022, doi:10.3390/biomedicines10081864_

Round 1
Reviewer 1 Report
This manuscripts details studies to test the effects of a diet enriched in lauric acid (SOLF) as a saturated fat vs oleic acid (UOLF) as an unsaturated fact in the ability of the adipokine leptin to signal in fat.
•It would be good to state a hypothesis going in, backed up by the work that led to a directional hypothesis.
•How were these fats -lauric vs oleic – chosen as the model for a saturated and unsaturated fat? How do these fats play a role in human diet?
•The components of standard chow should be shared, especially in terms of the fats already present.
•Did animals consume a similar amount of food and/or calories with these different diets? This should be shared as food intake is a possible contributor to outcomes.
•This study would benefit from a diagram/cartoon that shows the differences in leptin signaling in WAT when exposed to lauric vs oleic acid. There are many small, discrete pieces of data that could be better synthesized in a figure like this.
•The discussion could also be improved by asking and answering the “So what” question. Why is it important to know – in a BIG picture- the differences these two diets cause in leptin signalling? Extending that message as you interpret it from your data would help.
•I would encourage the authors to use color in figures, especially as differentiating SOLF, UOLF and SD.
•Were controls for endpoints run in Westerns in figure 1 (e.g. known AKT and pAKT)?
•Figure 3 was hard to follow because the ‘sc WAT’ and ‘Vis-Wat’ titles aren’t quite over all figures. Consider repeating them as headers in C and D of this figure.
Author Response
MANUSCRIPT: biomedicines-1816312
RESPONSE TO EDITOR AND REVIEWERS
Dear Editor,
Thank you very much for giving us the opportunity to resubmit a reviewed version of our manuscript. We would like to express our gratitude to the referees and yourself for the comments, that helped to substantially improve the manuscript. We think that we have addressed all the concerns raised and we hope that, after this revision, the manuscript will be acceptable for publication in this journal.
Our replies to the specific observations of the referees together with the list of changes made in the manuscript are presented below. The new additions have been highlighted in the manuscript.
Yours sincerely,
Beatriz Merino.
Reviewer 1
This manuscripts details studies to test the effects of a diet enriched in lauric acid (SOLF) as a saturated fat vs oleic acid (UOLF) as an unsaturated fact in the ability of the adipokine leptin to signal in fat.
- It would be good to state a hypothesis going in, backed up by the work that led to a directional hypothesis.
We have reformulated this part of the introduction to state our hypothesis (see lines 63-67).
- How were these fats -lauric vs oleic – chosen as the model for a saturated and unsaturated fat? How do these fats play a role in human diet?
SOLF and UOLF diets were chosen because they contain fats often included in human diets. Thus, high-oleic sunflower oil (UOLF) is a common component of processed foods containing “healthy” fat, such as bakery and pastry products, canned food, dips and salad dressings, etc. The SOLF diet contains palm kernel oil (lauric and palmitic acids are the main fatty acids present in this fat), a type of fat used in food industry to replace hydrogenated fat (which has been banned in many countries due to its detrimental effect on human health). It is interesting to note that palm kernel oil has a composition very close to that of coconut oil, which is now considered as a “superfood” by many dietary trends, on the basis of some of the metabolic properties of lauric acid (a main component of both coconut and palm kernel oil). By using these diets, we wanted to investigate the influence of an excessive intake of oleic and lauric/palmitic acid-enriched diets on the autoparacrine effect of leptin.
- The components of standard chow should be shared, especially in terms of the fats already present.
This information is included as Supplementary material in the new version of the MS (Table 1 Supplementary Material), and was already published in Plaza et al., Mol Nut Food Res 2019 doi.org/10.1002/mnfr.201900110.
Table 1: Diet composition
|
|
SD |
UOLF |
SOLF |
|
Energy density (kcal/g) |
3.1 |
5.3 |
5.3 |
|
Energy from proteins (%) |
27.2 |
9.9 |
9.8 |
|
Energy from carbohydrates (%) |
55.5 |
20.5 |
20.4 |
|
Energy from lipids (%) |
17.5 |
69.6 |
69.8 |
|
Total proteins (%) |
21.0 |
13.1 |
12.9 |
|
Total carbohydrates (%) |
42.9 |
27.1 |
26.9 |
|
Total lipids (%) |
6.0 |
40.8 |
41.2 |
|
Oleic acid (%) |
1.2 |
31.3 |
10.0 |
|
Palmitic acid (%) |
0.7 |
3.0 |
6.1 |
|
Lauric acid (%) |
- |
- |
19.8 |
|
Other fatty acids (%) |
4.17 |
6.13 |
5.21 |
- This study would benefit from a diagram/cartoon that shows the differences in leptin signaling in WAT when exposed to lauric vs oleic acid. There are many small, discrete pieces of data that could be better synthesized in a figure like this.
The graphical abstract has been modified.
- Did animals consume a similar amount of food and/or calories with these different diets? This should be shared as food intake is a possible contributor to outcomes.
These data have been already published (Plaza et al., Mol Nut Food Res 2019 doi.org/10.1002/mnfr.201900110), since some WAT samples used in the current study were obtained from mice used in previous research. In that paper we reported that SOLF and UOLF triggered an increase of body weight associated to the increase of energy intake. Nevertheless, energy efficiency (weight gain/kcal consumed) was similar in control, SOLF and UOLF mice, indicating that these diets do not appear to affect energy balance.
- The discussion could also be improved by asking and answering the “So what” question. Why is it important to know -in a BIG picture- the differences these two diets cause in leptin signalling? Extending that message as you interpret it from your data would help.
We have included a new paragraph in the discussion section to address this issue (lines 434-442):
“Our findings show that the intake of food enriched with either oleic (UOLF) or lauric/palmitic acids (SOLF) may differently affect the autoparacrine role of leptin. Overall, our findings suggest that SOLF desensitize signalling pathways downstream LepR in adipocytes, which could limit WAT adaptive mechanisms. Nevertheless, and despite UOLF does not alter LepR signalling, this diet trigger adipocyte hypertrophy in a similar extent than SOLF, indicating that, regardless of diet composition, an excessive intake of fat is harmful for WAT function. Further research aimed at correlating LepR responsiveness with adipocyte function in SOLF- and UOLF-fed mice appears necessary.”
- I would encourage the authors to use colour in figures, especially as differentiating SOLF, UOLF and SD.
We have modified the Figures as suggested.
- Were controls for endpoints run in Westerns in figure 1 (e.g. known AKT and pAKT)?
Phosphorylated Akt, STAT3 and AMPK were normalized with respect to total Akt, STAT3 and AMPK, respectively. Since total Akt, STAT3 and AMPK remained unchanged, we used these proteins as loading controls instead β-actin.
- Figure 3 was hard to follow because the ‘sc WAT’ and ‘Vis-Wat’ titles aren’t quite over all figures. Consider repeating them as headers in C and D of this figure.
Figure 3 has been modified as suggested.
Reviewer 2 Report
In the reviewed article, the authors show the effect of HFD containing a high content of lauric acid (SOLF) and HFD containing a high content of oleic acid (UOLF) on the autocrine / paracrine effect of leptin on adipose tissue in mice. The results presented are interesting, but I have many objections to this paper.
line 64, aim: ...influence of saturated and unsaturated fat on the functionality of LepR... Oleic acid belongs to monounsaturated fatty acids, while another group that has not been studied here is polyunsaturated fatty acids (PUFA) from the omega-3 and omega-6 groups. They can have a completely different effect than the 18:1 on LepR functionality. For this reason, the goal should be precisely formulated: ...the effect of saturated and MONOunsaturated fat on the functionality of LepR...
line 83: (see diets’ composition in Plaza et al., 2019b) - provide citation in Biomedicines format. Moreover, it should be at least stated here that the diet contains about 40% of energy from fat (18:1 or 12:0) comparing to 6% from fat in control diet. It would be best to measure FA composition in all types of chow using appropriate method - see below.
line 90: A part of the study (WAT composition and gene expression studies) carried out in samples from animals used in a previous published 91 study [12]. Does it mean that other results are fron another mice? Why WAT composition and gene expression wre not measured in the same animals?
Methods - WAT fatty acid composition was measured by 13C NMR. In my opinion it is not a best method to measure FA composition. Why FA was not assayed by HPLC as in chow in previos study of these authors (ref. 12)? The best method would be gas chromatography which allow to identify much more FA, that are present in WAT in lower amounts (eg. https://pubmed.ncbi.nlm.nih.gov/26678792/).
Methods: 2.4 leptin resistance assesment - Only the dose of leptin is provided. It is not explained how the leptin resistance was measured.
The description of the results of section 3.1 is unclear. It is not known which groups are compared to each other - comparison of UOLF and SOLF to control or UOLF to SOLF? What does dietary treatment mean - SOLF or UOLF? In figure 1, sign the SD SLF and UOLF under each panel (A-E). Similarly, in tables 3 and 4 in the diet column, it is not known which diet effect (SOLF or UOLF) are mentioned. These should be clarified and presented in diffrent way.
Table 5 - a few FA are included in the analysis. GC method would allow to present much wider FA profile. It is surprising, that linoleic acid is not detected in visceral WAT. Other studies showed similar content of this FA in visc and Sc WAT of mice (eg. https://pubmed.ncbi.nlm.nih.gov/26488808/; https://pubmed.ncbi.nlm.nih.gov/31788013/)
Discussion:
line 273 "diet enriched in saturated TG" - there is no data that saturated FA are present in TG fraction
line 277-278: "One of the most remarkable data of our study is the increase of pSTAT3, pAKT and 277 pAMPK observed specifically in SOLF mice, particularly in the Sc-WAT." It should be added that this effect was not dependent on leptin treatment.
Author Response
MANUSCRIPT: biomedicines-1816312
RESPONSE TO EDITOR AND REVIEWERS
Dear Editor,
Thank you very much for giving us the opportunity to resubmit a reviewed version of our manuscript. We would like to express our gratitude to the referees and yourself for the comments, that helped to substantially improve the manuscript. We think that we have addressed all the concerns raised and we hope that, after this revision, the manuscript will be acceptable for publication in this journal.
Our replies to the specific observations of the referees together with the list of changes made in the manuscript are presented below. The new additions have been highlighted in the manuscript.
Yours sincerely,
Beatriz Merino.
Reviewer 2
In the reviewed article, the authors show the effect of HFD containing a high content of lauric acid (SOLF) and HFD containing a high content of oleic acid (UOLF) on the autocrine/paracrine effect of leptin on adipose tissue in mice. The results presented are interesting, but I have many objections to this paper.
- line 64, aim: ...influence of saturated and unsaturated fat on the functionality of LepR... Oleic acid belongs to monounsaturated fatty acids, while another group that has not been studied here is polyunsaturated fatty acids (PUFA) from the omega-3 and omega-6 groups. They can have a completely different effect than the 18:1 on LepR functionality. For this reason, the goal should be precisely formulated: ...the effect of saturated and MONOunsaturated fat on the functionality of LepR...
We have reformulated this part of the introduction to state our hypothesis (see lines 63-67).
- line 83: (see diets’ composition in Plaza et al., 2019b) - provide citation in Biomedicines format. Moreover, it should be at least stated here that the diet contains about 40% of energy from fat (18:1 or 12:0) comparing to 6% from fat in control diet. It would be best to measure FA composition in all types of chow using appropriate method - see below.
Reference style has been adapted to Biomedicines format.
The paragraph 2.1 has been rewritten as suggested by the referee (lines 83-91):
“After one week, animals with similar average body weight (BW) were randomly assigned to one of three experimental groups that were fed (8 weeks) either a standard chow (SD, Teklad global 2018, Harlan Laboratories, IN, USA), SD enriched with 40% high-oleic sunflower oil (Unsaturated OiL-enriched Food, UOLF), or SD enriched with 40% palm kernel oil (Saturated OiL-enriched Food, SOLF) (see diets’ composition in [12]). UOLF and SOLF provide about 70 % energy from fat (vs 18 % in the case of chow), with oleic acid (18.1 cis 9; 31%) and lauric acid (12:0)/palmitic acid (16:0) (20%/6%) being the most abundant fatty acids in each diet. BW was monitored once a week.”
- line 90: A part of the study (WAT composition and gene expression studies) carried out in samples from animals used in a previous published 91 study [12]. Does it mean that other results are from another mice? Why WAT composition and gene expression were not measured in the same animals?
We have modified this part of 2.1, as suggested by the referee (lines 91-94):
“WAT morphology and composition and gene expression were analysed in samples obtained from animals used in a previous investigation [12]. A second group of mice was employed to assess leptin resistance in WAT’’.
- Methods - WAT fatty acid composition was measured by 13C NMR. In my opinion it is not a best method to measure FA composition. Why FA was not assayed by HPLC as in chow in previous study of these authors (ref. 12)? The best method would be gas chromatography which allow to identify much more FA, that are present in WAT in lower amounts (eg. https://pubmed.ncbi.nlm.nih.gov/26678792/).
We agree with the reviewer that GC is the most common and performant method for this kind of analysis. Nevertheless 13C-NMR is also considered a reliable method to analyse lipids in complex mixtures (Molecules, doi:10.3390/molecules22101663; MRC, doi.org/10.1002/mrc.2629; J Am Oil Chem Soc, DOI 10.1007/s11746-011-1848-2; Lipids, doi.org/10.1002/lipd.12277; J Pharm Biomed Anal, doi.org/10.1016/j.jpba.2022.114658…). We also agree that GC may yield a more detailed profile of fatty acid composition since it allows quantifying minor components. In our study we aimed at identifying to which extent food composition may condition WAT composition and this is the reason we decided to use 13C NMR, which is a technique currently running in our lab and performant enough to quantify major fatty acids in a mixture. Otherwise, this technique has been used by us in previous studies (Plaza et al., J. Endo. doi: 10.1530/JOE-17-0580.). We have now included a new paragraph in the Discussion section to clarify this issue (lines 388-391).
- Methods: 2.4 leptin resistance assessment - Only the dose of leptin is provided. It is not explained how the leptin resistance was measured.
Paragraph 2.4. has been modified as suggested (lines 139-148):
“To evaluate whether leptin signalling in WAT was modified by the dietary treatment, a second experiment, aimed at evaluating the effect of exogenous leptin on LepR responsiveness, was carried out after 8-wk of either SD, SOLF or UOLF diet. Mice were treated (i.p.) either with saline or 1 mg/kg mouse recombinant leptin (Sigma, St. Lou-is, MO) at 0900 h. After 60 min, animals were decapitated under isoflurane anaesthesia, blood collected in chilled EDTA-coated polypropylene tubes, and Sc- and Vis-WAT dissected and stored at -80°C until assay. Both tissues were prepared for western blot, as described below. The dose of leptin was chosen based on previous studies of our group [11]. This dose has been shown to provide plasma leptin levels of approx. 80 ng/ml [11], which fits into the range of severe hyperleptinemia.”
- The description of the results of section 3.1 is unclear. It is not known which groups are compared to each other - comparison of UOLF and SOLF to control or UOLF to SOLF? What does dietary treatment mean - SOLF or UOLF? In figure 1, sign the SD SLF and UOLF under each panel (A-E). Similarly, in tables 3 and 4 in the diet column, it is not known which diet effect (SOLF or UOLF) are mentioned. These should be clarified and presented in different way.
Section 3:1 has been rewritten accordingly to your suggestions:
“The effect of dietary treatment and acute leptin administration (1 mg/kg) on 705Tyr-STAT3, 473Ser-pAkt and 172Tyr-AMPK phosphorylation levels was evaluated both in Sc- and Vis-WAT.
Subcutaneous WAT: As illustrated in Figures 1A, B and C, ANOVA-2 (see F values in Table 3) revealed an effect of leptin on relative pSTAT3 (P<0.001), pAkt (P<0.001), and pAMPK levels (P<0.001), that was dependent on the type of diet consumed (P<0.05, P<0.05 and P<0.01, respectively). Post-hoc analyses indicated that leptin increased pSTAT3, pAkt and pAMPK in control and UOLF mice but not in SOLF animals.
Visceral WAT: In this tissue, a significant effect of leptin on pSTAT3 (see F values in Table 4) (P<0.001), pAkt (P<0.001) and pAMPK relative levels (P<0.001) was also detected. The leptin effect depended on the type of diet consumed in the case of pSTAT3 (P<0.05) and pAkt (P<0.01) but not for pAMPK. Further post-hoc analysis revealed an effect of leptin on pSTAT3 and pAkt in SD and UOLF animals.”
- Table 5 - a few FA are included in the analysis. GC method would allow to present much wider FA profile. It is surprising, that linoleic acid is not detected in visceral WAT. Other studies showed similar content of this FA in visc and Sc WAT of mice (eg. https://pubmed.ncbi.nlm.nih.gov/26488808/; https://pubmed.ncbi.nlm.nih.gov/31788013/)
This question has been partially addressed previously. Concerning the absence of linoleic acid in perigonadal WAT, differences in fatty acid (FA) content between fat pads have been identified in several studies (Metab Syndr Relat Disord, doi: 10.1089/met.2008.0056; Progress in Lipid Research, doi.org/10.1016/j.plipres.2008.03.003). To our knowledge, FA composition of mouse perigonadal WAT has not been until now reported. Even if this tissue is considered as a standard visceral pad, some studies have evidenced that it is endowed with relevant functional specificities in C57BL mice, since immune cells in this tissue are mainly restricted to macrophages (Diabetologia (2015) 58:1601–1609 DOI 10.1007/s00125-015-3594-8). The absence of linoleic acid may be related to this peculiarity, since Ω-6 FA have been shown to trigger inflammatory responses (Simopoulos et al. Nutrients 2016; doi.org/10.3390/nu8030128), and therefore linoleic acid may interfere with macrophage function (Hidalgo et al., Front Physiol. doi.org/10.3389/fphys.2021.66833). Otherwise, the absence of linoleic acid cannot be attributed to dietary treatments as it was absent in control perigonadal WAT.
Discussion:
- line 273 "diet enriched in saturated TG" - there is no data that saturated FA are present in TG fraction.
We have replaced the sentence in line 311, that now reads:
“In the current study we have shown that long-term consumption of diets enriched in saturated FA (SOLF) affects phosphorylation levels of protein kinases integral to LepR signalling pathways and LepR responsiveness to leptin in WAT. In contrast, unsaturated diets (UOLF) were devoid of these effects.”
- line 277-278: "One of the most remarkable data of our study is the increase of pSTAT3, pAKT and 277 pAMPK observed specifically in SOLF mice, particularly in the Sc-WAT." It should be added that this effect was not dependent on leptin treatment.
Following your suggestion, this sentence has been rephrased:
“One of the most remarkable data of our study is the increase of basal pSTAT3, pAkt and pAMPK observed specifically in SOLF mice, particularly in the Sc-WAT.”
Round 2
Reviewer 1 Report
The authors have responded to my comments - thank you.
Author Response
Reviewer 1
The authors have responded to my comments - thank you.
Thank you for your positive recommendations.

Reviewer 2 Report
The manuscript has been revised aacording to ma comments. However, for clarity of results presented in tables 3 and 4 the results of post-hoc tests could be presented in additional supplementary table.
Author Response
Reviewer 2
However, for clarity of results presented in tables 3 and 4 the results of post-hoc tests could be presented in additional supplementary table.
As suggested by the referee, an additional post-hoc analysis has been added in Suppl. Material.